# Peroxisome Proliferator-Activated Receptor-α (*PPARα*) Expression in a Clinical Population of Pakistani Patients with Type 2 Diabetes and Dyslipidemia

**DOI:** 10.3390/ijms231810847

**Published:** 2022-09-16

**Authors:** Maria Arif, Tanmoy Mondal, Asifa Majeed, Christopher A. Loffredo, Brent E. Korba, Somiranjan Ghosh

**Affiliations:** 1Department of Biochemistry and Molecular Biology, Army Medical College, National University of Medical Sciences, Rawalpindi 46000, Punjab, Pakistan; 2Department of Biology, Howard University, Washington, DC 20059, USA; 3Department of Oncology, Lombardi Comprehensive Cancer Center, Georgetown University, Washington, DC 20057, USA; 4Department of Microbiology & Immunology, Georgetown University, Washington, DC 20057, USA

**Keywords:** type 2 diabetes, diabetic dyslipidemia, serum lipid profile, PPARα gene

## Abstract

Poor glycemic control and dyslipidemia are hallmarks of type 2 diabetes mellitus (T2DM), which predispose to cardiovascular diseases. Peroxisome proliferator-activated receptor-α (*PPARα*) has been associated with atherosclerosis, but its role in T2DM is less clear. Previously, we studied PPARα expression levels in diabetics with and without dyslipidemia (DD). In this study we described the association with fasting blood glucose, HbA1c levels and lipid levels of the study population. Patient demography and biochemical data were collected from hospitals in Islamabad, Pakistan, and RT-PCR data of *PPARα* expression were retrieved from our previous study from the same cohort. We performed t-tests and regression analysis to evaluate the relationships between PPARα expression and demographic and clinical variables. As expected, body mass index and HbA1c were elevated in T2DM and DD patients compared to controls. Blood lipids (total cholesterol, triglycerides, LDL and HDL) were significantly higher in the DD group compared to the other two groups. In the T2DM and DD groups, the *PPARα* expression was not associated with any of the physical and biochemical parameters measured in this study. Expression of the *PPARα* gene was independent of blood lipids and glycemic control in this study. Further research is necessary to better understand the biological parameters of *PPARα* expression.

## 1. Introduction

Type 2 diabetes mellitus (T2DM) is a multifactorial metabolic disorder and has become a public health issue globally [1]. In Pakistan, 6.3 million people are affected by T2DM, which is projected to increase to 11.4 million by 2030 [2]. In 2021, it was estimated that the prevalence of T2DM is between 7.6 and 11% in different regions of Pakistan [3].

Poor glycemic control and dyslipidemia are hallmarks of T2DM. Dyslipidemia is defined as elevated levels of total cholesterol or low-density lipoprotein (LDL) or triglyceride level, or it may be defined as low levels of high-density lipoprotein (HDL) cholesterol [4]. In T2DM, the exchange of cholesterol between LDL and HDL under the influence of cholesterol ester transfer protein is often increased and results in a decrease in HDL cholesterol. Such dyslipidemia, in turn, can lead to macro-vascular complications and other cardiovascular diseases [5]. The frequency of dyslipidemia is estimated to be about 68.66% in T2DM [6].

Peroxisome proliferator-activated receptor-α (PPARα) has been of scientific interest for its potential role in atherosclerosis, due to its effects on HDL turnover [7]. The gene also exerts control of myocardial metabolism by transcriptionally regulating genes encoding enzymes involved in fatty acid and glucose utilization [8]. Furthermore, the *PPARα* gene regulates lipid profiles by regulating genes such as the apo-lipoprotein gene, the lipoprotein lipase (LPL) gene, the ATP binding cassette A1 (*ABCA1*) gene, and the scavenger receptor B1 [9] gene. Downregulation of these receptors and proteins leads to dyslipidemia. Finally, *PPARα* can trigger an upsurge in insulin sensitivity by consuming lipid stores in cells [10]. T2DM is a primarily a disease of hyperglycemia due to a deficiency of insulin’s many functions, but serum lipids are also strongly affected by insulin [5]. Regardless of insulin resistance or insufficiency, dyslipidemias are frequently observed in diabetic populations. The role of *PPARα* in lipid metabolism is well known but the involvement or etiology of *PPARα* in T2DM remains to be fully characterized.

In this study, we checked the status of the physical and biochemical parameters of DD participants compared to T2DM and control groups and examined associations of those variables with PPARα expression.

## 2. Results

### 2.1. Participant Characteristics

#### 2.1.1. Physical Parameters

We recruited a total of 295 participants, excluding those who were hypertensive, diagnosed with carcinoma, or had chronic diseases of the liver. Those patients who were on PPAR agonist medication, lipid-lowering therapy, insulin, or glucophage therapy were also excluded from the study. For the total participants, Table 1 lists the demographic and physical characteristics, comparing T2DM and DD groups with the controls. The mean age in the control, T2DM, and DD groups were43.26 ± 15.34, 52.38 ± 14.96, and 51.25 ± 10.27, respectively. The frequency of gender was 53 male and 47 female in the control group, 27 male and 68 female in the T2DM group, and 31 male and 69 female in the DD group (*p*-values 0.0005 and 0.0016 in T2DM and DD groups versus controls, respectively). Body weight (kg) and body mass index (BMI: kg/m^2^) of the participants of the T2DM *(p*-values 0.0233 and <0.0001) and DD (*p*-values 0.0334 and <0.0001) groups were significantly higher as compared to the control group.

#### 2.1.2. Fasting Blood Glucose and HbA1c Levels

As expected, the average HbA1c (%) and fasting blood glucose (mg/dl) levels were significantly (*p*-value < 0.0001) higher in the T2DM group (7.677 ± 1.326% and 190.83 ± 81.26 mg/dl respectively) as well as in the DD group (8.345 ± 2.325% and 228.40 ± 121.09 mg/dl respectively) compared to the control group (5.31 ± 0.89% and 86.54 ± 12.82 mg/dl respectively) (Figure 1). The highest levels of HbA1c (%) and fasting blood glucose (mg/dl) were observed in the DD group.

#### 2.1.3. Lipid Panel Levels

In the DD group, the average level of LDL was significantly higher (123.5 ± 33.17 mg/dl, *p*-value < 0.001) than in the control group (87.43 ± 22.99 mg/dl), whereas the amount of HDL in the DD group was significantly lower (35.61 ± 8.46 mg/dl, *p*-value < 0.001) (Figure 2A,B). We also observed a noticeable difference in total cholesterol (mg/dl) and triglycerides (mg/dl) content between the control and DD groups (*p*-value 0.001), whereas there were no statistically significant different differences between controls and T2DM in these parameters (Figure 2C,D).

### 2.2. Correlations among HbAc1, Fasting Blood Glucose, and Blood Lipids Panel

A correlation analysis was carried out to see if any biological parameters were linked to the diabetic or dyslipidemic status (Figure 3A–E and Figure 4A–D). We did not observe any significant correlations between HbA1c levels and LDL, HDL, total cholesterol, and triglycerides in the T2DM group of participants (Figure 3A,C,E and Figure 4C). In the DD group, the HbA1c level was significantly correlated with total cholesterol (*p*-value < 0.0001), fasting blood glucose (*p*-value < 0.0001) and triglycerides (*p*-value = 0.003) (Figure 4B,D).

### 2.3. PPARα Gene Expression

The mean ± SD of PPARα expression data were retrieved from our previous manuscript [11], which were 20.48 ± 1.587, 26.15 ± 1.701, and 29.43 ± 3.263 in the control, T2DM, and DD groups, respectively. The coefficient of variation (%) was 7.750, 6.504, and 11.09 in the control, T2DM, and DD groups, respectively. Compared to the control group, the Ct mean of the DM and DD levels was higher (or downregulated in terms of relative expression ratio) and statistically significant (*p*-value < 0.0001) (Figure 5).

#### Associations of PPARα Gene Expression with Age, HbA1c, Fasting Blood Glucose, and Lipid Panel

We did not observe any statistically significant associations between PPARα gene expression levels and the physical biochemical parameters of interest in this study (e.g., age, fasting blood glucose, HbA1c, and BMI) neither in controls nor within the T2DM and DD groups (Appendix A). We also did not observe any significant correlations with lipid parameters such as total cholesterol, triglycerides, LDL, and HDL (Figure 6 and Figure 7). We checked for any associations even after adjusting for gender, but none were found to be statistically significant (Appendix A) except fasting blood glucose and HbA1c levels: both parameters were significantly correlated with *PPARα* in the male control group (*p*-values 0.004 and 0.035 respectively), and only fasting blood glucose was correlated in the T2DM male group (*p*-value 0.016).

## 3. Discussion

In the present study, we evaluated the associations of *PPARα* expression with physical and biochemical parameters of T2DM and DD patients compared to controls. The expression of *PPARα* was independent of any physical or biochemical characteristics. As expected, all of the measured biochemical parameters (e.g., fasting blood glucose, HbA1c, LDL, total cholesterol, and triglycerides) were higher in the DD group than in the other groups, and the levels of triglycerides and total cholesterol in the DD group were correlated with the HbA1c level of those individuals.

A rising trend of dyslipidemia has been observed globally and is a major public health concern [12]. Dyslipidemia is also highly prevalent in the Pakistani population [13], with estimates as high as 63% [14]. T2DM is primarily a disease of hyperglycemia due to a deficiency of insulin’s many functions, but serum lipids are also strongly affected by insulin. Dyslipidemias are commonly seen in diabetic populations irrespective of insulin deficiency or resistance [15]. Lipid abnormalities may be the result of the unbalanced metabolic state of T2DM, and improved control of hyperglycemia does moderate diabetes-associated dyslipidemia [16]. We observed that there was a considerable elevation in triglycerides and total cholesterol in the DD group, which is consistent with other reports in Pakistani populations [2,17], and that these levels were correlated with increasing HbA1c levels. A similar observation was reported in an Afghani population with T2DM [18].

According to recent epidemiological data, dyslipidemia is on the rise in urban regions of Pakistan, probably because of a higher socioeconomic status, a sedentary lifestyle, poor eating habits, and a lack of physical activity [14]. The rapid urbanization with improved economic status over time is predicted to exacerbate these trends [19], particularly in older individuals who are most at risk for developing the disorder [20].

*PPARα* is crucial in the regulation of inflammation and angiogenesis [21] and is expressed in the liver, kidney, heart, muscle, adipose tissue, and other organs with significant fatty-acid catabolism [22]. Defects in PPARs have been linked to lipodystrophy, obesity, and insulin resistance as a result of the impairment of adipose tissue expandability and functionality [23]. Similarly, obesity causes systemic insulin resistance, which raises the levels of free fatty acids in the blood and is also related to dyslipidemia [23]. A typical dyslipidemia pattern in T2DM includes elevated serum triglycerides, decreased high-density lipoprotein cholesterol, and, occasionally, elevated low-density lipoprotein cholesterol levels [24]. According to recent data, medicines that lower blood sugar levels can also affect lipid profiles by either directly or indirectly modulating the expression of the PPARα gene [24,25,26,27,28]. Since all the individuals in our study had newly been diagnosed with diabetes, we did not consider the use of diabetic medication as an influential factor.

PPARα also controls the expression of a wide range of hepatic genes encoding for proteins involved in fatty acid catabolism and lipoprotein metabolism. Its activation leads to changes in the transcription of multiple genes that regulate lipid and lipoprotein metabolism including *LPL*, *APOC3*, *APOA1*, and *APOA5* [23]. Finally, clinical studies suggest that PPARα is associated not only with triglycerides and LDL but also with inflammation marker responses to fenofibrate intervention [24]. These outcomes could be caused by a variety of other variables, including environmental and epigenetic change, which require further, in-depth investigation. Although we were able to link *PPARα* expression with the diabetic dyslipidemia condition itself, compared to T2DM with normal lipids and in comparison to controls, none of the measured lipid values such as HDL and LDL were associated with PPARα expression, in contrast to what other studies have reported [11,16]. 

Our study had some weaknesses, in addition to its strengths of having recruited a moderately large sample of nearly 300 participants from the Rawalpindi area of Pakistan and detecting consistent relationships between high HbA1c levels and abnormal blood lipid panels within the disease groups. On the other hand, PPARα expression was not associated with physical and biochemical parameters, which may in part be due to the unequal gender and age distributions among the groups we studied.

## 4. Materials and Methods

### 4.1. Study Ethics and Approval

The study was conducted after approval by the Ethical Review Committee of the Army Medical College, National University of Medical Sciences, Pakistan (No. ERC/MS-17 dated 11 August 2017), which provided approval of recruitment efforts at Fouji Foundation Hospital and the Military Hospital in Rawalpindi. Signed informed consent was obtained from all participants.

### 4.2. Study Participants

The eligible participants were recruited by physicians between 2018 and 2020. A total of 295 participants were sequentially enrolled, including those who volunteered for the control group (*n* = 100) without having a diagnosis of T2DM or DD conditions; the T2DM group (*n* = 95); and DD group (*n* = 100). The T2DM participants were diagnosed according to the American Diabetes Association [29] guideline (HbA1c level of 5.7% or higher), and the diagnosis criteria for the DD group were based on the National Cholesterol Education Program [30]: >200 mg/dl for total cholesterol and >50 mg/dl for HDL (Appendix A).

### 4.3. Sample Collection and Biochemical Levels

Participants donated a total of 7 mL of blood samples that were collected in ethylene diamine tetra acetic acid (EDTA) (3 mL), Sodium Fluoride (2 mL) and serum gel tubes (2 mL). Fasting blood glucose level was detected by Glucose Assay Kit (Cat. No. STA-680 Cell Biolabs Inc., San Diego, CA, USA). HbA1c was measured using the BioMajesty, HbA1c kit method (JCA-BM6010/C; Cat. No. 1334899 10964 DiaSys Diagnostic Systems GmbH, Alte Strasse, Holzheim, Germany). Lipid profile was assessed with commercial assay kits, e.g., blood levels of triglycerides were measured by Triglyceride (TG) Colorimetric Assay Kit (Cat. No. E-BC-K238 Elabscience, Houston, Texas, USA). HDL-cholesterol was quantified using the HDL-Cholesterol Differential Precipitation Enzymatic colorimetric test (Cat. No. 1133010 Atlas Medical, Cambridge, UK). LDL-cholesterol was quantified by the formula; LDL-C = TC − [HDL-C + TG/5)]. Serum total cholesterol was measured using the Total Cholesterol Assay Kit (Cat. No. STA-384 Cell Biolabs Inc., San Diego, CA, USA). Samples were analyzed and absorbance was measured using the Chemistry Analyzer SystexChemix 180.

### 4.4. RNA Isolation, cDNA Synthesis, and Quantitative Real-Time Polymerase Chain Reaction (qRT-PCR)

All the blood samples were collected in EDTA tubes (3 mL) (BD Biosciences, Farnklin Lakes, NJ, USA) for qRT-PCR. RNA extraction was carried out using the GeneJET RNA Purification Kit (Cat. No. K0732 ThermoFisher Scientific Inc., Waltham, MA, USA) from whole blood cells. The quantitative analysis including the purity and concentration of the RNA was checked by using the Spectrophotometer NanoDrop One (Thermo Fisher Scientific Inc., Waltham, MA, USA) according to the manufacturer’s protocol. Gel electrophoresis was employed for the qualitative analysis of RNA samples. cDNA synthesis was carried out using the RevertAid First Strand cDNA Synthesis Kit as per the manufacturer’s recommended protocol (Cat. No.: K1621, ThermoFisher Scientific Inc., Waltham, MA, USA). Quantitative PCR was performed on ABI 7500 real Time PCR using Maxima SYBR Green /ROXqPCR Master Mix (Cat. No. K0221 by ThermoFisher Scientific Inc., Waltham, MA, USA). Primer design and all details of the procedure for *PPARα* expression were mentioned in our previous manuscript [11].

### 4.5. Statistical Data Analysis

Data were analyzed using GraphPad Prism (version 8) software. All measurements are expressed as mean ± SD for continuous variables. The Chi-square test was used to compare proportions between the groups, and Student’s *t* test was used to compare pairwise groups for continuous variables. Association between the outcome variable of *PPARα* gene expression and various independent factors was performed by using linear regression analysis, adjusting for age and gender.

## 5. Conclusions

In this case-control study, we confirmed the high levels of fasting blood glucose, HbA1c, and dysregulated lipids in T2DM and DD patients in Pakistan in relation to controls. The expression of *PPARα* appeared to be independent of physical and biochemical parameters. Further research is necessary to better understand the biological associations of *PPARα* expression in T2DM.

## Figures and Tables

**Figure 1 ijms-23-10847-f001:**
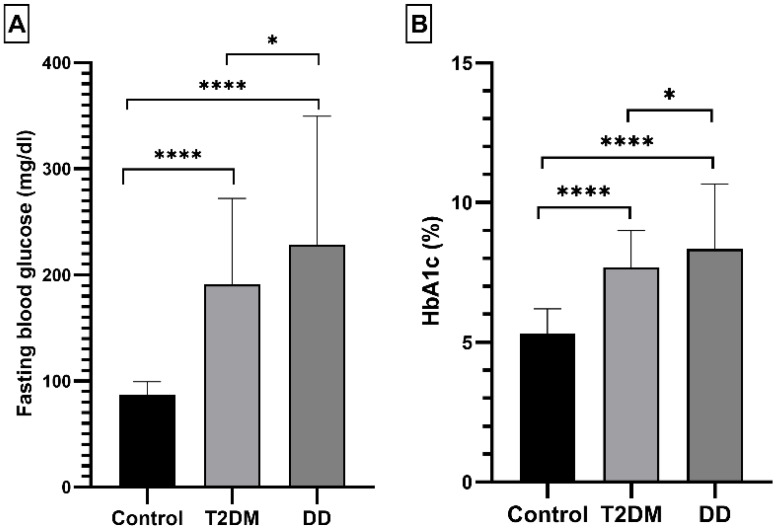
The differences in HbA1c (%) and fasting blood glucose between the control, T2DM, and DD groups. Each bar represents the group’s mean. Error bars are ± standard deviation. Note: T2DM and DD data are compared with the control group. * and **** denote statistical significance (*p* < 0.01; *p* < 0.0001 respectively). (**A**): Comparison of BSF (mg/dl), (**B**); Comparison of HbA1c (mmol/mol).

**Figure 2 ijms-23-10847-f002:**
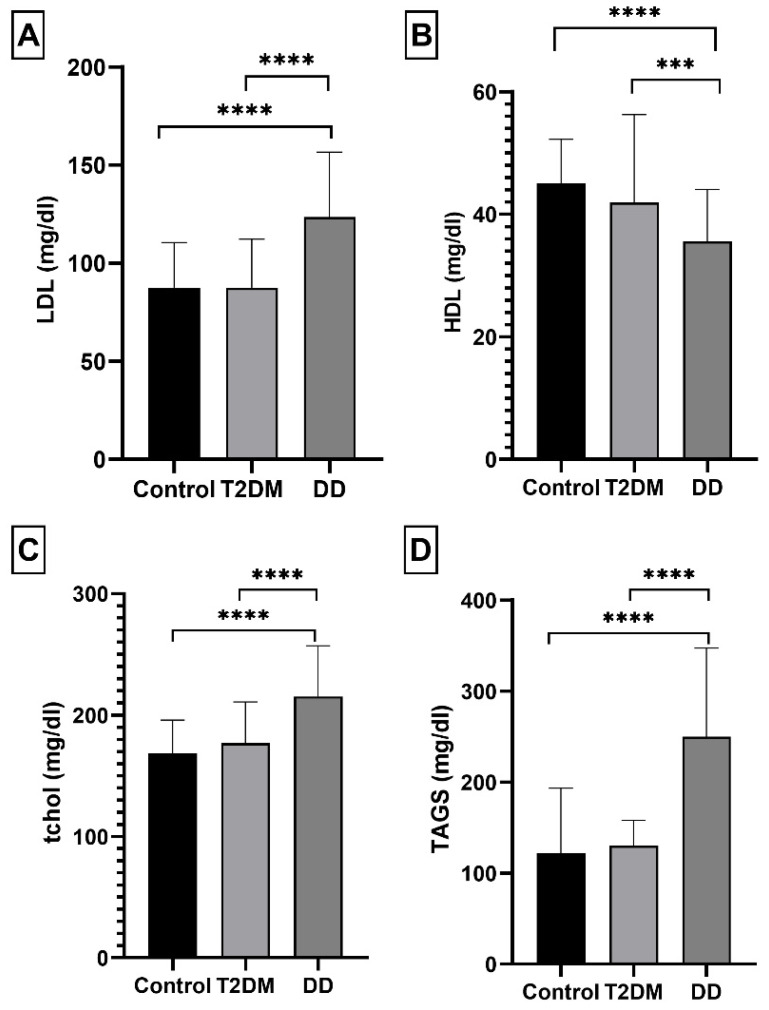
The differences in parameters between the control, T2DM, and DD groups. Each bar represents the group’s mean. Error bars are ± standard deviation Note: T2DM and DD data are compared with the control group. *** and **** denote statistically significant (*p* < 0.001; *p* < 0.0001 respectively). (**A**): Low density lipoprotein (LDL mg/dl); (**B**): High-density lipoprotein (HDL mg/dl); (**C**): Total serum cholesterol (tchol mg/dl); (**D**): Serum triglycerides (TAGS mg/dl).

**Figure 3 ijms-23-10847-f003:**
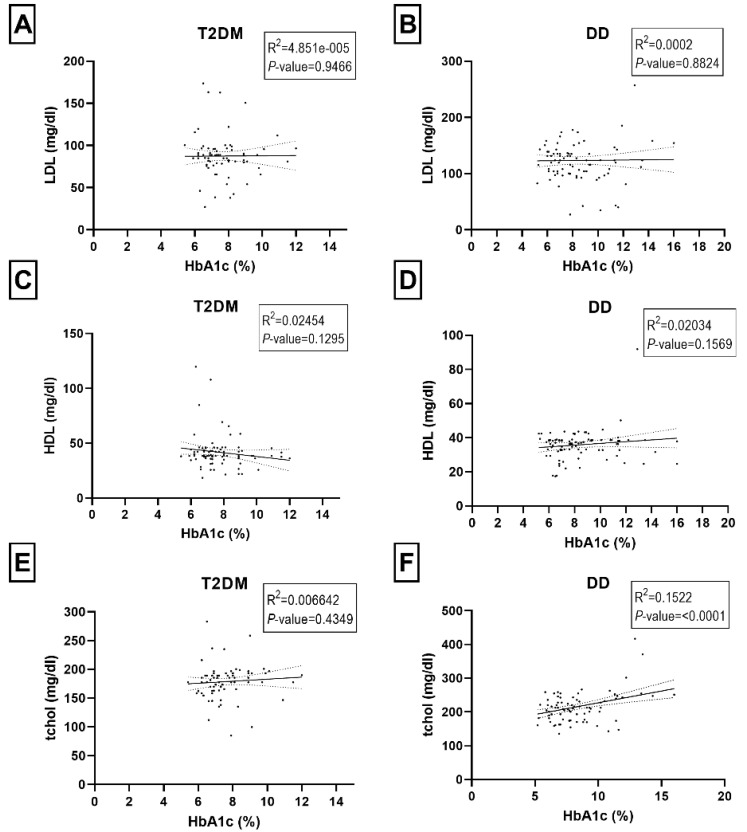
Relationship between HbA1C (%) and biomedical parameters of low-density lipoprotein (LDL mg/dl) (**A**,**B**), high-density lipoprotein (HDL mg/dl) (**C**,**D**), and total serum cholesterol (tchol mg/dl) (**E**,**F**) in T2DM and DD group. Each dot of the scattered plot represents the data of individual participants.

**Figure 4 ijms-23-10847-f004:**
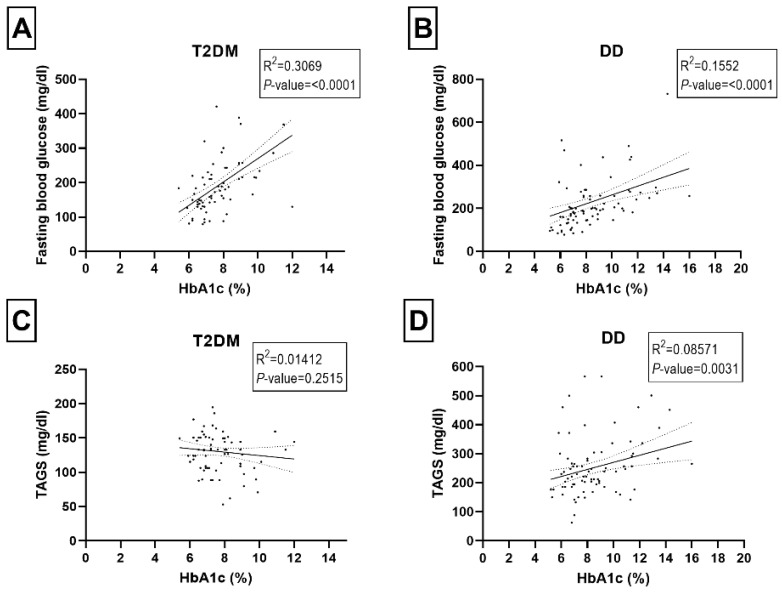
Relationship between HbA1C (%) and biomedical parameters of fasting blood glucose (mg/dl) (**A**,**B**), serum triglycerides (TAGS mg/dl) (**C**,**D**) in T2DM and DD group. Each dot of the scattered plot represents the data of individual participants.

**Figure 5 ijms-23-10847-f005:**
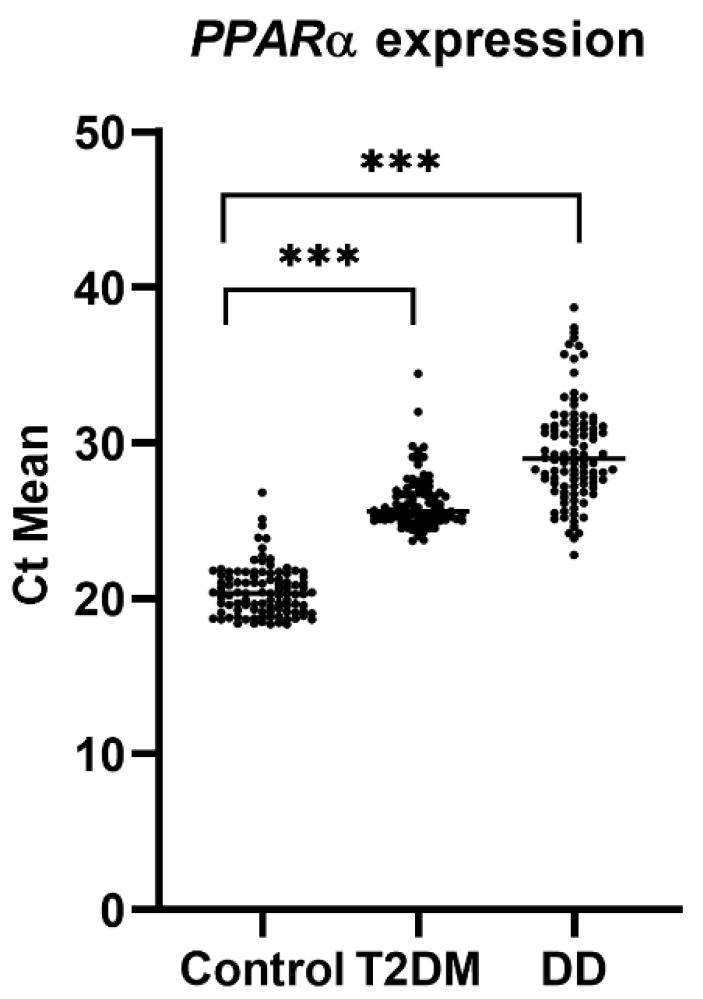
The mean Ct value of PPARα expression in the control, T2DM, and DD groups. Each black dot indicates the mean Ct value of an individual sample. *** denote statistical significance (*p* < 0.001). Data were collected from our previous study. (Arif M, Siddique K, Majeed A. Comparison of expression of PPARα and SCARB1 gene in type 2 diabetic and type 2 diabetic dyslipidemia patients. *Medicina*. (Submitted and under review; Manuscript ID medicina-1845712).

**Figure 6 ijms-23-10847-f006:**
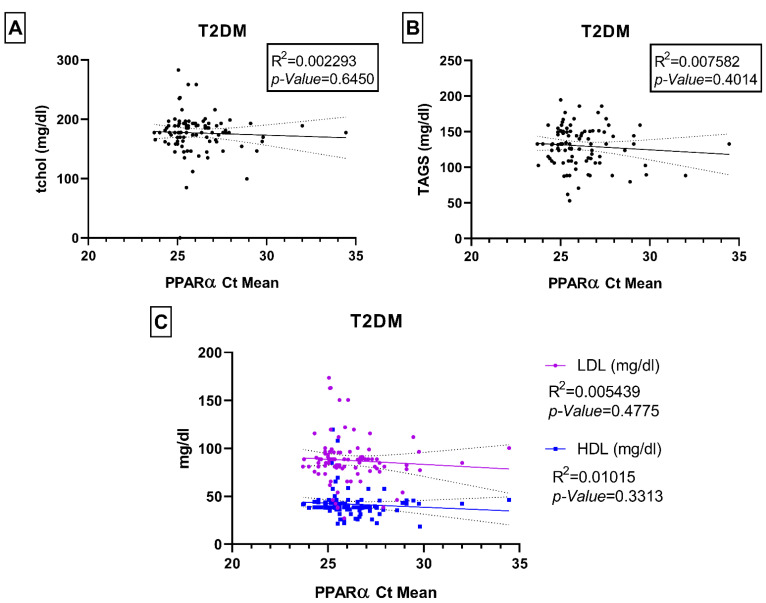
Relationship between PPARα Ct mean and biomedical parameters of total serum cholesterol (tchol mg/dl) (**A**), serum triglycerides (TAGS mg/dl) (**B**), low-density lipoprotein (LDL mg/dl) and high-density lipoprotein (HDL mg/dl) (**C**) in T2DM group. Each dot of the scattered plot represents the data of individual participants.

**Figure 7 ijms-23-10847-f007:**
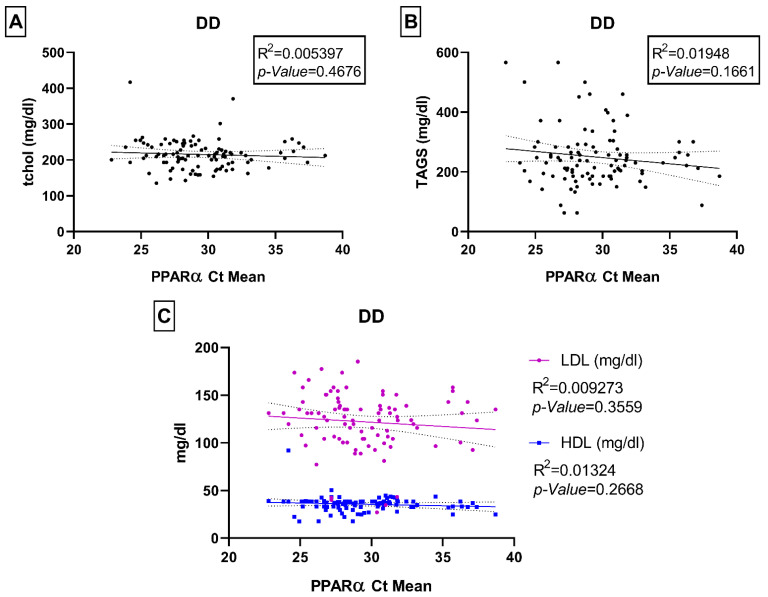
Relationship between PPARα Ct mean and biomedical parameters of total serum cholesterol (tchol mg/dl) (**A**), serum triglycerides (TAGS mg/dl) (**B**), low-density lipoprotein (LDL mg/dl) and high-density lipoprotein (HDL mg/dl) (**C**) in DD group. Each dot of the scattered plot represents the data of individual participants.

**Table 1 ijms-23-10847-t001:** Characteristics of the cohort.

Characteristics of the Patients	Control (n = 100)	T2DM (n = 95)	*p*-Value of T2DM Compared to Control	DD (n = 100)	*p*-Value of DD Compared to Control	*p*-Value T2DM vs. DD
**Age (Y)**	43.26 ± 15.34	52.38 ± 14.96	<0.0001	51.25 ± 10.27	<0.0001	0.53
**Gender**	M-53; F-47	M-27; F-68	0.0005	M-31; F-69	0.0016	0.69
**Weight (kg)**	67.81 ± 9.54	71.02 ± 10.00	0.0233	70.67 ± 9.34	0.0334	0.80
**BMI (kg/m^2^)**	23.83 ± 4.04	26.81 ± 4.42	<0.0001	25.89 ± 3.16	<0.0001	0.53

## Data Availability

Not applicable.

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
