# Peer review of "Peroxisome Proliferator-Activated Receptor-α (PPARα) Expression in a Clinical Population of Pakistani Patients with Type 2 Diabetes and Dyslipidemia"

_ijms, 2022, doi:10.3390/ijms231810847_

Round 1
Reviewer 1 Report
The authors focused on Peroxisome proliferator-activated receptor-α (PPARα) expression in a clinical population of Pakistani patients with type 2 diabetes and dyslipidemia. Please see my suggestions bellow:
The purpose of the study is not enough specified in the Introduction. What is the novelty/special aspects that your research brings to the field?
The methodology of the study is a little bit ambiguous. You state that ”PPARα expression was retrieved from our previous study”, the reader can understand that the variable of some patients such as lipid parameters etc were correlated with data of other patients. Please clarify.
The results are not very clearly presented, you correlate hba1c with lipid parameters, but you do not show the statistical association between PPAR and all the parameters, even if they do not correlate in some associations this data must be shown.
Table 1. Head of the Table, first cell is empty. It is not allowed for a scientific paper. ”Characteristics of the patients” would be a good choice or something similar.
Be consistent with denotation. For “kg/m2” use 2 at superscript. Revise the entire manuscript in this regard (ie. L68”)
Extend all figures on the entire width of the page, as the MDPI draft allows it. They will be readable, as in the actual form the written is too small.
The Discussion chapter must be improved. Please discuss the use of lipid lowering medication in down-regulation of PPAR. Please check: https://www.mdpi.com/2075-4418/10/7/483 . Please discuss the relationship between insulin resistance and PPAR https://pubmed.ncbi.nlm.nih.gov/33101480/. The main results of the study which I find important, although you were able to link 171 PPARα expression with the diabetic dyslipidaemia condition itself, is not very well valorised. A figure that explains the design of the study and the main outcomes would be helpful.
Author Response
Reviewer 1 comments and response:
The authors focused on Peroxisome proliferator-activated receptor-α (PPARα) expression in a clinical population of Pakistani patients with type 2 diabetes and dyslipidemia. Please see my suggestions bellow:
- The purpose of the study is not enough specified in the Introduction. What is the novelty/special aspects that your research brings to the field?
Response: We thank to the reviewer for the advice. The introduction section is now updated with the special aspects of our study.
- The methodology of the study is a little bit ambiguous. You state that ”PPARα expression was retrieved from our previous study”, the reader can understand that the variable of some patients such as lipid parameters etc were correlated with data of other patients. Please clarify.
Response: We appreciate the reviewer for bringing out the point. The population of PPARα expression and the population with all the recorded lipid parameters were the same clinical population. We updated the text in the methodology section to clarify this point.
- The results are not very clearly presented, you correlate hba1c with lipid parameters, but you do not show the statistical association between PPAR and all the parameters, even if they do not correlate in some associations this data must be shown.
Response: We thank reviewers for the recommendation. We added additional two figures related to the association between PPARα Ct mean and lipid profile in the T2DM and DD groups.
- Table 1. Head of the Table, first cell is empty. It is not allowed for a scientific paper. ”Characteristics of the patients” would be a good choice or something similar.
Response: We thank to the reviewer for the suggestion. We updated the table as per the recommendation.
- Be consistent with denotation. For “kg/m2” use 2 at superscript. Revise the entire manuscript in this regard (ie. L68”)
Response: The denotation “kg/m2’ updated to kg/m2 as advised.
- Extend all figures on the entire width of the page, as the MDPI draft allows it. They will be readable, as in the actual form the written is too small.
Response: As per the advice of the reviewers, we updated all the figures on the entire width of the page to make it large and readable.
- The Discussion chapter must be improved. Please discuss the use of lipid lowering medication in down-regulation of PPAR. Please check: https://www.mdpi.com/2075-4418/10/7/483 . Please discuss the relationship between insulin resistance and PPAR https://pubmed.ncbi.nlm.nih.gov/33101480/. The main results of the study which I find important, although you were able to link 171 PPARα expression with the diabetic dyslipidaemia condition itself, is not very well valorised. A figure that explains the design of the study and the main outcomes would be helpful.
Response: We appreciate the reviewer for bringing out the point and offering the relevant citation. The discussion section is now updated with the information regarding the relationship between lipid lowering medication and PPAR expression, and we have also added some information about the relationship between insulin resistance and PPAR expression.
Reviewer 2 Report
In the manuscript “Peroxisome proliferator-activated receptor-α (PPARα) expression in a clinical population of Pakistani patients with type 2 diabetes and dyslipidemia” Arif et al provide a detailed analysis of metabolic data for three relatively large groups of patients, collected from hospitals in Islamabad (Pakistan): the control group, patients with Diabetes, and patients with both Diabetes and Dyslipidemia. This analysis largely matches commonly accepted concepts/findings. The authors further include the data on PPAR alpha expression and find no correlation of those with any other reported key metabolic parameters. This study is highly descriptive, but is potentially interesting regarding the increasing interest in using PPAR agonists in metabolic disorders.
Below are the minor points that need to be addressed:
1) The authors need to carefully reassess the way of presenting their data: there are numerous discrepancies/inaccuracies between the text and the figures. Below are some examples:
-Table 1. P values for DM/control, “weight” =0.02, “BMI”<0.0001; but, in the text, the values are 0.0233 and <0.001 respectively;
-Figure 1. The values (26.81; 25.89; 23.83) are reported as a fasting glucose level, mg/dl. However, the graph shows much higher, likely to be correct values;
-Figure 2B. The error bars seem too high for reported p values.
2) Figure 5. A better explanation of the data needs to be provided (shown are Ct values, those can be easily transformed into relative expression levels). The description in the manuscript (line 117: “DM and DD levels were higher and statistically significant”) is ambiguous: a reader can conclude that DM and DD groups have higher PPAR levels, while the opposite is the case.
3) How the samples (to measure PPAR alpha expression) were collected? This needs to be explained in the text.
4) I recommend to move the exclusion criteria (from Materials/Methods) to the main text, section 2.1.1, where the patients are first introduced. Regarding the exclusion criteria, it might be helpful to add a sentence specifying PPAR agonists.
5) Regarding correlation analyses for PPAR alpha: it might be useful to include an sentence on potential “age plus gender” dependencies.
Author Response
Reviewer 2 comments and response:
In the manuscript “Peroxisome proliferator-activated receptor-α (PPARα) expression in a clinical population of Pakistani patients with type 2 diabetes and dyslipidemia” Arif et al provide a detailed analysis of metabolic data for three relatively large groups of patients, collected from hospitals in Islamabad (Pakistan): the control group, patients with Diabetes, and patients with both Diabetes and Dyslipidemia. This analysis largely matches commonly accepted concepts/findings. The authors further include the data on PPAR alpha expression and find no correlation of those with any other reported key metabolic parameters. This study is highly descriptive but is potentially interesting regarding the increasing interest in using PPAR agonists in metabolic disorders.
Below are the minor points that need to be addressed:
- The authors need to carefully reassess the way of presenting their data: there are numerous discrepancies/inaccuracies between the text and the figures. Below are some examples:
-Table 1. P values for DM/control, “weight” =0.02, “BMI”<0.0001; but, in the text, the values are 0.0233 and <0.001 respectively;
-Figure 1. The values (26.81; 25.89; 23.83) are reported as a fasting glucose level, mg/dl. However, the graph shows much higher, likely to be correct values;
-Figure 2B. The error bars seem too high for reported p values.
Response: We are very much thankful to the reviewers for identifying the unintentional error. We corrected all the tables and figures and updated them in the manuscript.
- Figure 5. A better explanation of the data needs to be provided (shown are Ct values, those can be easily transformed into relative expression levels). The description in the manuscript (line 117: “DM and DD levels were higher and statistically significant”) is ambiguous: a reader can conclude that DM and DD groups have higher PPAR levels, while the opposite is the case.
Response: We agree to the reviewers for raising the issue. We revised the section with proper terminology in the updated manuscript.
- How the samples (to measure PPAR alpha expression) were collected? This needs to be explained in the text.
Response: We thank to the reviewer for the suggestions. We updated the manuscript with the details of the sample collection to measure the PPARa expression. The procedure in brief, is as follows. All the blood samples were collected in EDTA tubes (3ml) for qRT-PCR. The quantitative analysis including the purity and concentration of the RNA was checked by using the Spectrophotometer NanoDrop One (Thermo Fisher Scientific, MA, USA) according to manufacturer’s protocol. Gel electrophoresis was employed for the qualitative analysis of RNA samples.
- I recommend to move the exclusion criteria (from Materials/Methods) to the main text, section 2.1.1, where the patients are first introduced. Regarding the exclusion criteria, it might be helpful to add a sentence specifying PPAR agonists.
Response: We moved the exclusion criteria section from the Materials/Methods to the main text (section 2.1.1), as per the reviewer’s recommendation. Additionally, we added the PPAR agonist medication group as one of the exclusion criteria for our study in the 2.1.1 section.
- Regarding correlation analyses for PPAR alpha: it might be useful to include an sentence on potential “age plus gender” dependencies.
Response: We had already mentioned the age and gender correlations with PPARa expression in the main text (section of 2.3.1.) and all the corresponding figures were also introduced in the supplementary data file. We did not observe any statistically significant associations between PPARα gene expression levels and the physical parameters like age (Supplementary Figure file). We also checked for any associations even after adjusting for gender, but none were found to be statistically significant (Supplemental Table 3) except fasting blood glucose and HbA1c levels: both parameters were significantly correlated with PPARα in the male control group (p-values 0.004 and 0.035 respectively), and only fasting blood glucose was correlated in the T2DM male group (p-value 0.016).
Round 2
Reviewer 1 Report
All the requirements have been fulfilled. i recommend publication.